# The Neolignan Honokiol and Its Synthetic Derivative Honokiol Hexafluoro Reduce Neuroinflammation and Cellular Senescence in Microglia Cells

**DOI:** 10.3390/cells13191652

**Published:** 2024-10-04

**Authors:** Chiara Sasia, Vittoria Borgonetti, Caterina Mancini, Giulia Lori, Jack L. Arbiser, Maria Letizia Taddei, Nicoletta Galeotti

**Affiliations:** 1Department of Neurosciences, Psychology, Drug Research and Child Health (Neurofarba), University of Floence, Viale G. Pieraccini 6, 50121 Florence, Italy; chiara.sasia@unifi.it (C.S.); vittoria.borgonetti@unifi.it (V.B.); 2Department of Experimental and Clinical Medicine, University of Florence, Viale Morgagni 50, 50134 Florence, Italygiulia.lori@unifi.it (G.L.); 3Department of Dermatology, Emory School of Medicine, Winship Cancer Institute, Atlanta, GA 30322, USA; jarbise@emory.edu

**Keywords:** Honokiol, Claisened Hexafluoro, microglia, cellular senescence, neuroinflammation, Notch signaling

## Abstract

Microglia-mediated neuroinflammation has been linked to neurodegenerative disorders. Inflammation and aging contribute to microglial senescence. Microglial senescence promotes the development of neurodegenerative disorders, including Alzheimer’s disease (AD). In this study, we investigated the anti-neuroinflammatory and anti-senescence activity of Honokiol (HNK), a polyphenolic neolignane from *Magnolia officinalis* Rehder & E.H Wilson, in comparison with its synthetic analogue Honokiol Hexafluoro (CH). HNK reduced the pro-inflammatory cell morphology of LPS-stimulated BV2 microglia cells and increased the expression of the anti-inflammatory cytokine IL-10 with an efficacy comparable to CH. HNK and CH were also able to attenuate the alterations in cell morphology associated with cellular senescence in BV2 cells intermittently stimulated with LPS and significantly reduce the activity and expression of the senescence marker ß-galactosidase and the expression of p21 and pERK1/2. The treatments reduced the expression of senescence-associated secretory phenotype (SASP) factors IL-1ß and NF-kB, decreased ROS production, and abolished H2AX over phosphorylation (γ-H2AX) and acetylated H3 overexpression. Senescent microglia cells showed an increased expression of the Notch ligand Jagged1 that was reduced by HNK and CH with a comparable efficacy to the Notch inhibitor DAPT. Overall, our data illustrate a protective activity of HNK and CH on neuroinflammation and cellular senescence in microglia cells involving a Notch-signaling-mediated mechanism and suggesting a potential therapeutic contribution in aging-related neurodegenerative diseases.

## 1. Introduction

Aging is the general decline in functional capabilities of an organism over time [1]. A main hallmark of aging is the accumulation of senescent cells in different tissues, including the brain, that can negatively affect organ functionality and promote tumorigenesis [2]. Indeed, aging is frequently associated with the development of a huge number of pathological conditions defined as age-related diseases with a consequent shortening of lifespan. Growing evidence associates the onset and progression of neurodegenerative diseases with the acquisition of a senescent phenotype by brain cells, including microglia [3,4].

Inflammation and aging are significant modifiers of microglial functions [5]. Microglia cells are involved in the maintenance of homeostasis, protection against pathogens, and disorders of the CNS, and they contribute to neuronal development [6]. When stimulation becomes uncontrolled and constant, microglia cells lose their physiological functions and acquire a “senescent” phenotype [7]. Senescent cells are characterized by cell cycle arrest, a phenomenon first investigated as an antitumor intervention [8]. The acquisition of a pro-inflammatory phenotype, known as a senescence-associated secretory phenotype (SASP), is characterized by the release of SASP factors, including pro-inflammatory cytokines and reactive oxygen species (ROS) [1]. While an irreversible cell cycle withdrawal is relatively harmless, the pro-inflammatory molecules released by SASP activity can negatively affect the surrounding tissues, including neurons, contributing to tissue dysfunction and the development of pathological conditions [9]. Targeting senescent cells with specific senotherapeutics represents a promising therapeutic approach [10,11] and justifies the efforts to identify new strategies to dampen microglia activation [4]. 

In the research of new senotherapeutics, natural phenolic compounds, usually endowed with high efficacy and tolerability, represented an interesting perspective. Since senescent cells propagate systemic inflammation through the activation of pro-oxidant and pro-inflammatory pathways, polyphenols, molecules endowed with antioxidant and anti-inflammatory activity, might hold promise as an “anti-senescence” approach. Honokiol (HNK) is a polyphenolic neolignane extract from the bark of *Magnolia officinalis* Rehder & E.H Wilson, known and used since ancient times for its antioxidant, anti-inflammatory, neuroprotective, antidepressant, cardioprotective, antidiabetic, and anticancer properties [12]. Recent studies have shown in vitro and in vivo antitumor activity of HNK [13], at least in part related to the inhibition of the NF-kB pathway, which is associated with the activation of genes implicated in many types of hematologic and solid neoplasia [14] and mediates tumor cell proliferation, inhibits apoptosis, and induces angiogenesis [15]. This promising anticancer activity led to the identification of HNK derivatives with the aim of enhancing its therapeutic activity; among them, Honokiol Hexafluoro (also known as Claisened Hexafluoro; CH) has been shown to be effective against metastatic melanoma [16] and to inhibit DUX4-induced toxicity in an in vitro model of facioscapulohumeral muscular dystrophy (FSHD) [17]. 

HNK can cross the blood–brain barrier, and it has been shown to have an important neuroprotective action [18]. During cerebral ischemia–reperfusion activities, HNK has been shown to dampen inflammation by reducing cytokine production, NF-κB stimulation [19], and ROS production [20]. Through its antioxidant and anti-neuroinflammatory activity, it showed neuroprotective action in the toxicity of β-amyloid (Aβ) [21]. Recently, through the attenuation of astrocyte and microglia activation, CH neuroprotective activity was described in a mouse model of HIV infection [22]. Inflammation and cellular senescence are closely related to neurological disorders [23]. Thus, the aim of this work was to investigate the effect of HNK and its fully synthetic derivative CH in reducing microglial senescence to evaluate their potential clinical value in neurodegenerative disorders.

## 2. Materials and Methods

### 2.1. BV2 Cell Culture 

BV2 murine immortalized microglial cells (mouse, C57BL/6, brain, microglial cells, Tema Ricerca, Genova, Italy; 16–20 passages) were used. The cells were thawed and kept in culture in a 75 cm^2^ flask (Sarstedt, Milano, Italy) in a medium containing RPMI with the addition of 10% of heat-inactivated (56 °C, 30 min) fetal bovine serum (FBS, Gibco^®^, Milan, Italy) 1% glutamine, and a 1% penicillin–streptomycin solution (Merck, Darmstadt, Germany). Cells were cultured at 37 °C and 5% CO_2_ with daily medium change until confluence (70–80%). Trypan blue staining was used for cell count.

To induce neuroinflammation and microglial senescence, BV2 cells were stimulated with a bacterial lipopolysaccharide from Gram- (LPS, Merck, Darmstadt, Germany). Briefly, for the neuroinflammation model, BV2 cells were treated for 24 h with LPS 250 ng/mL in minimal medium (RPMI with 3% FBS). For the senescence model, BV2 cells were treated 4 times for 4 h/day for a total of 10 days with LPS 500 ng/mL [24]. This is a well-established model for microglial senescence.

### 2.2. Cell Treatment

HNK (Sigma Aldrich, Milan, Italy) was dissolved in 1 mL of pure DMSO and solubilized in RPMI to reach the final concentrations of 0.1, 1, 3, and 10 µM (0.5% DMSO). CH (4,4′-(Hexafluoropropane-2,2-diyl)bis(2-allylphenol)) was provided by Dr. Arbiser. It was 99% pure as assessed by nuclear magnetic resonance and was of pharmaceutical grade. CH was dissolved in pure DMSO to obtain a 10 mM stock solution and solubilized in RPMI to reach the final concentrations. DAPT (3 µM; Sigma Aldrich, Milan, Italy) was dissolved in 0.5% DMSO. Rosmarinic acid (RA; Merck, Darmstadt, Germany) was solubilized directly in RPMI cell culture medium at a concentration of 1 mg/mL, filtered (Filter syringe 0.2 µm, 30 mm, Biosigma, Venice, Italy), and then diluted in the medium to obtain final concentrations of 1 µM. Cells were incubated with tested substances for 24 h.

### 2.3. Sulforhodamine B (SRB) Assay

Cell viability was assessed by the sulforhodamine B (SRB) assay [25]. Cells were seeded in 96-well plates (2.0 × 10^4^ cells per well in 200 μL). After treatment, cells were fixed in 50% trichloroacetic acid (TCA, Merck, Darmstadt, Germany) in RPMI at 4 °C for 1 h. Then, they were treated with SRB 4 mg/mL in acetic acid 1% and incubated for 30 min at RT. Then, wells were washed four times with 1% acetic acid (200 µL per well), and 200 µL of TRIS HCl solution (pH 10) was added to the wells and incubated for 5 min with shaking. The absorbance was finally recorded using a multiplate reader at a wavelength of 570 nm. The treatments were performed in 6 replicates in 3 independent experiments, and cell viability was calculated by normalizing the values to the mean of the control.

### 2.4. Senescence-Associated Heterochromatin Foci Analysis (SAHFs)

After treatments, BV2 cells were fixed with 4% PFA for 30 min at 4 °C. Following incubation with blocking buffer (PBS, containing 1% bovine serum albumin) for 2 h at RT, primary antibodies towards AcH3 (1:200 in PBSA 5%; Santa Cruz Biotechnology, Santa Cruz, CA, USA, Cat#sc-56616) were added for 2 h at RT. The primary antibodies were then removed, and fixed cells were incubated in secondary antibodies labeled with Invitrogen Alexa Fluor 568 (578–603, 1:500; Thermo Fisher Scientific, Walthan, MA, USA) for 1 h. A solution of DAPI in mounting medium (90% glycerol + PBS) was added, and images were acquired (OLYMPUS BX63F fluorescence microscope). Values were normalized to control. Three independent experiments were performed. 

### 2.5. Senescence-Associated ß-Galactosidase Activity Assay

To evaluate ß-galactosidase activity, an ONPG kit (Merck, Darmstadt, Germany) was used following the manufacturer’s instructions. Three independent experiments were performed.

### 2.6. Senescence-Associated β-Galactosidase Staining

Cells were fixed with 3% paraformaldehyde in PBS for 5 min and then incubated at 37 °C in a non-humidified incubator under atmospheric CO_2_ conditions for 16 h in a freshly prepared senescence-associated β-galactosidase (SA-β-Gal) staining solution as previously described [26]. Three independent experiments were carried out to evaluate the effects of treatments.

### 2.7. Protein Lysates from Cells

Cells (3 × 10^5^) were seeded in 6-well plates, and after treatment, proteins from cells were extracted by RIPA buffer (50 mM Tris-HCl pH 7.4, 150 mM NaCl 1% sodium deoxycholate, 1% Triton X-100, 2 mM PMSF) (Merck, Milan, Italy). The insoluble pellet was separated by centrifugation (12,000× *g* for 30 min, 4 °C). The total protein concentration in the supernatant was measured using the Bradford colorimetric method (Merck).

For cell fractionation, cytoplasmic and nuclear lysates were separated by resuspending cells in a hypotonic cold solution (10 mM HEPES pH 8, 10 mM KCl, 1.5 mM MgCl2, 1 mM DTT, 0.1 mM EDTA, 0.2% NP40). The whole-cell lysate was then centrifuged at 16,000× *g* for 10 min at 4 °C ensuring the separation of the two fractions. Extracted proteins were then quantified.

### 2.8. Western Blot (WB)

Protein samples (30 µg of protein/lane) were separated by 10% SDS-polyacrylamide gel electrophoresis (SDS-PAGE) and transferred to nitrocellulose membranes for 90 min at 110 V using standard procedures [27]. After blocking, membranes were incubated overnight at 4 °C with primary antibodies against ß-galactosidase, NFκB p65, IL-10, JAGGED-1, p21, AcH3, β-Actin (Santa Cruz Biotechnology), H2AX, phospho-H2AX, IkBα, and p44/42 MAPK (Erk1/2) (Cell Signaling, Danvers, MS, USA). HRP-conjugated antibodies were used as secondary antibodies. Band intensity was detected through chemiluminescence (Chemidoc, BIO-RAD, Milan, Italy) and quantified using ImageJ 2.14 (NIH). For each measure, the signal intensity was normalized to total protein content or β-Actin. Three independent experiments (*n* = 3) were conducted.

### 2.9. Real-Time PCR (RT-PCR)

Total RNA was extracted with the RNeasy kit (Qiagen, Hilden, Germany) according to the manufacturer’s instructions. The RNA concentration and quality were determined using Nanodrop ONE (Thermo Fisher Scientific). Total RNA in the amount of 1000 ng was reverse-transcribed with the QuantiTect Reverse Transcription Kit (Qiagen #205311) according to the manufacturer’s instructions. mRNA expression levels were determined by Real-Time PCR using a QuantiFast SYBR Green Kit (Qiagen #204054). The nucleotide sequences of the specific primers (Thermo Fisher Scientific) used are reported in the Table 1 below.

Amplifications were run on a CFX96 C1000 Touch Real-Time System (Bio-Rad). Data were normalized on β2 microglobulin (β2M). Three independent experiments were performed.

### 2.10. Total ROS Production 

Cells were incubated with 5 μM 2′,7′-dichlorodihydrofluorescein diacetate (H2DCF-DA) (Sigma Aldrich) at 37 °C for 10 min. Cells were then lysed with 500 μL RIPA buffer and centrifuged at 14,000 rpm for 1 min. An amount of 100 μL of cell lysates was transferred to a 96-well plate, and fluorescence was measured at a wavelength excitation/emission of 585 nm/535 nm using a Biotek Synergy H1 microplate reader. The results were normalized on total protein content quantified with the BCA assay. Three independent experiments were performed.

### 2.11. Cell Morphology Analysis

Cell morphological analysis was performed by experimenters blind to the cell culture conditions. Images were taken by a Nikon ECLIPSE TS2-S-SM optical microscope and analyzed through the ImageJ program. Cells were counted per mm^2^ microscopic area in at least 10 randomly selected fields. For each treatment group, three independent experiments were performed. The average diameter and length of cellular processes along with the soma surface area (average measurements from 120 individual cells) were evaluated. Cells were then grouped into three different categories: (a) small (<200 µm^2^), (b) mid-sized (200–400 µm^2^), and (c) large (>400 µm^2^) cells. 

### 2.12. Statistical Analysis

The results are expressed as mean ± SEM. A one-way and two-way analysis of variance (ANOVA) followed by the Tukey or Bonferroni post hoc test, respectively, were used for statistical analysis of the in vitro data. Student’s t test was used when necessary. Values of *p* < 0.05 were considered significant. The software GraphPad Prism version 10.1.0 (GraphPad Software, San Diego, CA, USA) was used in all statistical analyses.

## 3. Results

### 3.1. Effect of HNK and CH in an In Vitro Model of Neuroinflammation

The dose–response effect on cell viability induced by HNK and CH was investigated on BV2 microglia cells. Both treatments, at doses ranging from 1 to 10 µM, never reduced cell viability (Figure 1A). Thus, in the following experiment, doses of 0.1, 1, 3, and 10 µM were used. We evaluated the effect of both compounds on LPS-stimulated BV2 cells used as a model of neuroinflammation. Exposure to LPS (250 ng/mL for 24 h) induced a pro-inflammatory state that drastically reduced cell viability (Figure 1B,C). This effect was prevented by both HNK (3 µM) and CH (10 µM) (Figure 1B). The treatments also reduced the number of cells in the pro-inflammatory state, with a peak effect at 10 µM. The treatments showed an efficacy comparable to that of rosmarinic acid (RA), used as a senomorphic reference drug [24] (Figure 1A,B).

LPS stimulation induced a morphological change in microglia cells. Resting BV2 cells were mainly found in a short, round morphology, while LPS caused a stretched and elongated phenotype. HNK (3–10 µM) and CH (10 µM) reduced the number of cells in the elongated pro-inflammatory phenotype (Figure 1D,E). LPS increased the BV2 cell surface area. This effect was attenuated by HNK, while CH showed a trend towards surface area reduction without reaching statistical significance (Figure 1F). By performing a more detailed morphological characterization, we evaluated the redistribution of the cell subpopulation by surface area in each treatment group. In the unstimulated BV2 group, most cells were small (<200 µm^2^) or medium-sized (200–400 µm^2^) (59.2% and 30%, respectively), and only a small amount were in the big-sized group (>400 µm^2^) (10.8%). Under LPS stimulation, the small-sized cell number was drastically reduced (16%), and a shift towards the medium-sized and large cells was induced (43 and 41%, respectively). Both treatments reduced the number of larger cells, and HNK also increased the number of small-sized cells (Figure 1G,H), showing an anti-inflammatory effect. 

After stimulation with LPS 250 ng/mL, a reduction in the protein expression of the anti-inflammatory cytokine IL-10 was observed, confirming the presence of a pro-inflammatory activation process. Treatments with HNK and CH at all concentrations tested increased IL-10 expression, further supporting the induction of an anti-inflammatory effect (Figure 1I).

### 3.2. Effect of HNK and CH in an In Vitro Model of Cellular Senescence

#### 3.2.1. Effect of HNK and CH on Altered Cell Morphology of Senescent BV Cells

Positive results on neuroinflammation encouraged us to investigate the capability to attenuate microglia cellular senescence by the investigated compounds. Intermittent stimulation of BV2 cells with LPS (500 ng/mL) for 10 days (Figure 2A) induced a condition of cellular senescence, as previously reported [24,28]. Senescent cells showed a morphological change characterized by the acquisition of an oblong and ramified phenotype. BV2 cells intermittently exposed to LPS acquired a predominant pro-inflammatory phenotype that was reduced by both HNK and CH with a peak effect at 1 and 3 µM (Figure 2B). Senescent cells showed a pronounced increase in the cell surface area that was reduced by both treatments at all doses tested (Figure 2C,D). A more detailed morphological analysis showed a significantly different distribution by size of senescent cells in comparison with control, unstimulated cells (Figure 2E). Approximately 60% of the cells were large-sized, 30% were medium-sized, and 10% were small-sized (Figure 2F), showing a different pattern of distribution from that of cells in the pro-inflammatory state (Figure 1H) characterized by a prevalence of large-sized cells. HNK and CH attenuated the senescence-associated morphology by drastically reducing the number of large-sized cells to CTRL levels (Figure 2E,F).

#### 3.2.2. Effect of CH and HNK on Senescence-Associated Markers

The main molecular biomarkers of senescent cells include senescence-associated β-galactosidase (SA-β-gal) and p53/p21. To further characterize the anti-senescence profile of HNK and CH, the effects produced on these markers was investigated.

A characteristic trait of a senescent cell is the increased activity and protein expression of SA-β-gal. Upon intermittent stimulation with LPS, the BV2 cells showed a robust enhancement of both SA-β-gal protein expression (Figure 3A) and activity (Figure 3B). Treatment with HNK and CH at all the concentrations tested was effective in reducing both parameters (Figure 3A,B). Quantification of SA-β-gal staining further confirmed the increased SA-β-gal activity in intact cells as well as the capability of CH and HNK to restore activity to control values. The efficacy was comparable to that of rosmarinic acid (RA), used as reference drug (Figure 3C).

The hallmarks of senescent cells comprise, other than increased SA-β-gal, the activation of the p53/p21 pathway with consequent cell cycle arrest [29]. The levels of p21 were, therefore, investigated. As shown in Figure 3D, p21 expression increases following LPS stimulation, while HNK and CH hinder this effect. Similar results were observed for the expression of p44/42 ERK (Figure 3E), key drivers of the senescent phenotype [30]: HNK hinders the LPS-induced increase in p44/p42 expression, while CH results as less effective.

#### 3.2.3. Anti-SASP Activity of CH and HNK

Senescent cells, metabolically active even though cell-cycle arrested, acquire a senescence-associated secretory phenotype (SASP) characterized by the production of interleukins, inflammatory cytokines, and growth factors able to affect the behavior of the surrounding cells [31]. In line with the acquisition of a senescent phenotype, LPS-treated BV2 cells increase the expression of *IL-6* and *IL-1β* (Figure 4 A,B), two main SASP-associated factors, while HNK and CH counteract the production of these interleukins, confirming their senomorphic activity.

NF-kB is the main driver of both microglial activation and SASP activity. Upon inflammatory stimulation, it translocates from the cytosol to the nucleus and promotes the transcription of pro-inflammatory genes. Senescent cells showed a significant decrease in the protein expression of cytosolic NF-kB, indicating an activation of the NF-kB signaling. NF-kB activation was attenuated in a CH dose-dependent manner, as demonstrated by the increased levels of cytosolic NF-kB, with significant effects at 3 and 10 µM. HNK increased cytosolic NF-kB levels at all concentrations tested (Figure 4 C). In agreement, CH and HNK stimulation reduces the increased expression of IKBα2 promoted by LPS stimulation (Figure 4D).

Increased reactive oxygen species (ROS) and diffuse DNA damage contribute to the acquisition of a senescent phenotype [32]. In keeping with this evidence, LPS induces an increase in ROS content and higher levels of DNA damage, including the deposition of γ-H2AX, a biomarker of DNA double-strand breaks involved in the recruitment of DNA repair complexes. On the contrary, HNK and CH treatment restores ROS almost to control levels and decreases the phosphorylation levels of H2AX (Figure 4E,F), endorsing the antioxidative role of HNK and CH, in agreement with previously reported data for HNK [33]. Acetylated Histone 3 (acH3) expression increases following the onset of cellular senescence [28]. This was also observed in our experiment, where in senescent cells higher levels of acH3 were detected by immunofluorescence investigation (Figure 4G,H) and Western blotting (Figure 4I). CH and HNK treatment dose-dependently reduced the acH3 overexpression, with a peak effect at 3 µM (Figure 4G–I).

### 3.3. Effect of CH and HNK on Jagged-1 Expression in BV2 Senescent Cells

Experimental evidence has shown a role of the Notch signaling pathway in microglia activation [34] as well as in cellular senescence [35], where it appears to have a main role in the paracrine mechanism that induces senescence in surrounding cells [36]. An increase in Jagged-1 protein expression was observed in senescent BV2 cells in comparison with unstimulated cells (Figure 5A,B). HNK and CH, at the highest effective doses in attenuating cellular senescence (3 µM), restore Jagged-1 basal levels with an efficacy comparable to DAPT, a γ-secretase inhibitor used as a Notch signaling inhibitor reference drug (Figure 5C). HNK and CH (3 µM) reduced SA-ß-gal activity (Figure 5D), the cell surface area (Figure 5E), and the number of cells in the pro-inflammatory state (Figure 5F,G) with an efficacy comparable to DAPT, indicating a potential involvement of the Notch signaling pathway in the CH and HNK anti-senescence mechanism.

## 4. Discussion

Even though the anti-inflammatory activity of HNK has already been observed [18], little is known about its anti-neuroinflammatory and anti-senescence effects at the microglial level. We found that HNK was able to reduce the pro-inflammatory phenotype of LPS-stimulated BV2 murine microglial cells, a model of neuroinflammation, and to increase the level of the anti-inflammatory mediator IL-10. The activity of HNK was compared to that of its synthetic derivative CH, showing a comparable anti-inflammatory profile and similar efficacy. 

Intermittent LPS stimulation of BV2 cells for 10 days has been proposed as an in vitro model of microglia senescence. These cells showed increased sized and irregular shape, morphological alterations that have been described as one of the hallmarks of microglia senescence [37,38]. Exposure to intermittent LPS stimulation caused a shift of cells towards a large-sized phenotype (approximately 60%) at the expenses of small-sized cells that reduced from 70% in the untreated cells to 15%. Treatment with HNK and CH drastically reduced the subpopulation of large-sized cells up to control values and partially rescued the subpopulation of small-sized cells. However, to assess the development of cellular senescence, it is recommended to detect a combination of cellular markers, prominently senescence-associated alterations in protein expression [39]. The induction of a senescent phenotype on microglia cells was further supported by the observation of increased activity and expression of SA-ß-galactosidase (SA-ß-gal), p21, and pERK1/2. SA-ß-gal activity detectable at pH 6.0 is widely recognized as the most common marker of cellular senescence. Indeed, the maximal enzymatic activity of SA-ß-gal is at pH 4.0, whereas at pH 6.0 the enzymatic activity is reduced by almost 99%, and only senescent cells with augmented lysosomal content possess enough enzyme to display activity [40]. A typical feature of cellular senescence is the cell cycle arrest that is mainly mediated by the p53/p21 pathway [41]. Consistently, an upregulation of p21 protein expression was observed in microglia cells. Finally, an overexpression of pERK1/2, a cellular pathway reported to be associated with senescence [42], was detected. HNK and CH, in addition to reducing the size and number of cells in the activated status, reduced senescence markers with comparable efficacy, showing the capability to attenuate microglia cellular senescence.

The acquisition of an SASP phenotype is a well-known feature of cellular senescence. This phenomenon mainly consists of the secretion of different chemokines, pro-inflammatory cytokines, growth factors, and matrix-remodeling enzymes [43,44] and represents one of the main detrimental characteristics of the aging process and increased susceptibility to the development of age-related diseases. The SASP response is regulated at the transcriptional level mainly through the CCAAT/enhancer-binding protein β (C/EBP-β) and nuclear factor kappa-B (NF-κB) transcription factors [45,46]. Upon activation, NF-κB translocates into the nucleus and transactivates several SASP genes [31]. NF-κB activity is reinforced by the increased production of IL-1 [46] that, in turn, contributes to the upregulation of IL-6 and IL-8 [47]. HNK and CH reduced the overexpression of pro-inflammatory markers, such as IL-6 and IL-1β, and attenuated the activation of the NF-κB signaling pathway. Considering the pivotal role of NF-κB in the promotion of SASP, the inhibition of this pathway in senescent cells has been reported to selectively repress SASP genes [44]. Thus, the senotherapeutic properties of HNK and CH suggest a potential contribution to reducing age-related disease, such as heart disease, diabetes, and atherosclerosis. Some evidence to support this hypothesis has been provided by preclinical studies that reported a beneficial effect of HNK on type 2 diabetes mellitus [48].

Recently, a potent neuroprotective effect of HNK has been reported in multiple animal models of central nervous system diseases, including cerebrovascular injury, spinal cord injury, anxiety, epilepsy, and cognitive disorders, most likely due to its activity in inhibiting oxidative stress and neuronal excitotoxicity, alleviating neuroinflammation, and regulating mitochondrial function [12,49,50]. Indeed, HNK and CH enhance SIRT3 expression, thus preventing mitochondrial and cytosolic ROS accumulation [51]. Recently, Balasubramaniam et al. [52] demonstrated that SIRT3 is essential in mediating the HNK and CH suppression of palmitate and the LPS-induced mitochondrial oxidative stress in enteric neurons. These data highlight the neuroprotective effect of HNK and CH and suggest their potential use as therapeutics in neurodegenerative disorders. In addition, senescence-associated heterochromatic foci (SAHFs) are also characteristic changes to chromatin that can be detected as a DNA/chromatin-dense foci in senescent cells [53]. SAHFs are mainly formed in fibroblasts but can also be present also in other cell types, such as microglia, astrocytes, and melanocytes [54]. In microglia, SAHFs are enriched for typical heterochromatin marks, such as acH3K9 [28]. Consistently, we found and increased SAHFs in microglia senescent cells that were attenuated by both HNK and CH.

The Notch signaling pathway is involved in cell-to-cell interactions between adjacent cells and consists of four receptor subtypes (Notch1, 2, 3, and 4), expressed on signal-receiving cells, and their ligands (Delta-like 1, 3, and 4 and Jagged 1 and 2), expressed on signal-sending cells [55]. The role of Notch signaling in tumors has been widely described, but several studies showed the contribution of Notch1 to cellular senescence by the regulation of SASP [56]. SASP acts in a paracrine fashion to induce secondary senescence in surrounding cells [57], a process that involves Notch1 signaling [35]. A specific type of senescence induced by Notch has been described [58], characterized by a specific upregulation of the ligand Jagged1 that, by engagement of NOTCH1, drives senescence on adjacent cells through the modulation of chromatin structure [59]. Several reports indicated that HNK involves the Notch signaling pathway in its anticancer activity [60,61], and recent findings described the HNK modulation of the Notch receptor Jagged1 in a model of peripheral neuropathy [62]. We observed an increased expression of the Jagged1 protein in microglial senescent cells that was attenuated by HNK and CH treatment with an efficacy profile comparable to the γ-secretase inhibitor DAPT, a widely used Notch inhibitor reference drug [63].

## 5. Conclusions

In this study, we demonstrated the anti-inflammatory and senolytic effects of HNK and CH on microglia cells. Our findings suggest a protective role on microglia-mediated neuroinflammation. Microglial senescence is a well-known contributor to neurodegenerative disease but has not yet been addressed as a therapeutic target. Current FDA-approved therapies have shown modest effects in slowing neurodegeneration, and targeting microglial senescence offers an additional therapeutic target. Effective prevention and treatment of neurodegenerative disorders will likely require a combination of therapies rather that a single agent, and compounds like HNK and CH may be complementary to established therapies in combating neurodegeneration.

## Figures and Tables

**Figure 1 cells-13-01652-f001:**
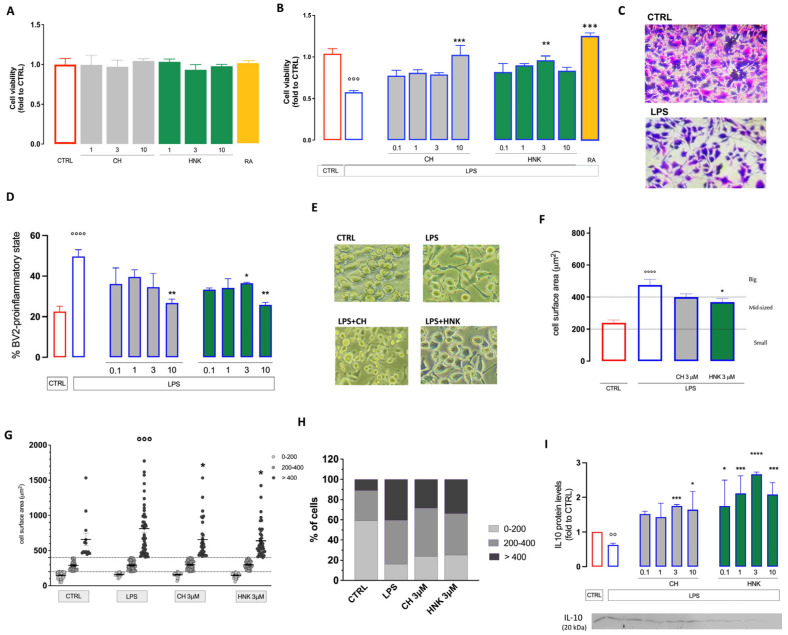
Effect of CH in an in vitro model of neuroinflammation. (**A**) Lack of effect of cell viability by CH (1–10 µM), HNK (1–10 µM), and the reference drug rosmarinic acid (RA; 1 µM) on BV2 cells using the SRB colorimetric assay. (**B**) Reversal by CH and HNK treatment (0.1–10 µM) of the reduction in cell viability induced by LPS 250 ng/mL. RA (1 µM) was used as reference drug. (**C**) Representative images of CTRL and LPS-stimulated BV2 cells (SRB assay), bar 30 μm. (**D**) Reduction in the number of BV2 cells in the pro-inflammatory state after LPS stimulation by CH and HNK (10 µM). (**E**) Representative images of CH- and HNK-treated LPS-stimulated BV2 cells. (**F**) CH and HNK (10 µM) reduced the increase in the cell surface induced by LPS stimulation, bar 20 μm. (**G**) Redistribution of BV2 cell subpopulation by cell surface area after 24 h LPS stimulation in the presence or absence of CH and HNK. Cells were classified as either small (<200 µm^2^), mid-sized (200–400 µm^2^), or large (>400 µm^2^). (**H**) Effect of CH and HNK on LPS-induced variation of the percentage of distribution of small, mid-sized, and big cells. (**I**) Reduction in protein expression of the anti-inflammatory cytokine IL10 by LPS and reversal by treatments with CH and HNK (10 µM). Three independent experiments were carried out to evaluate the effects of treatments. * *p* < 0.05, ** *p* < 0.01, *** *p* < 0.001, **** *p* < 0.0001 vs. LPS; °° *p* < 0.01, °°° *p* < 0.001, °°°° *p* < 0.0001 vs. CTRL.

**Figure 2 cells-13-01652-f002:**
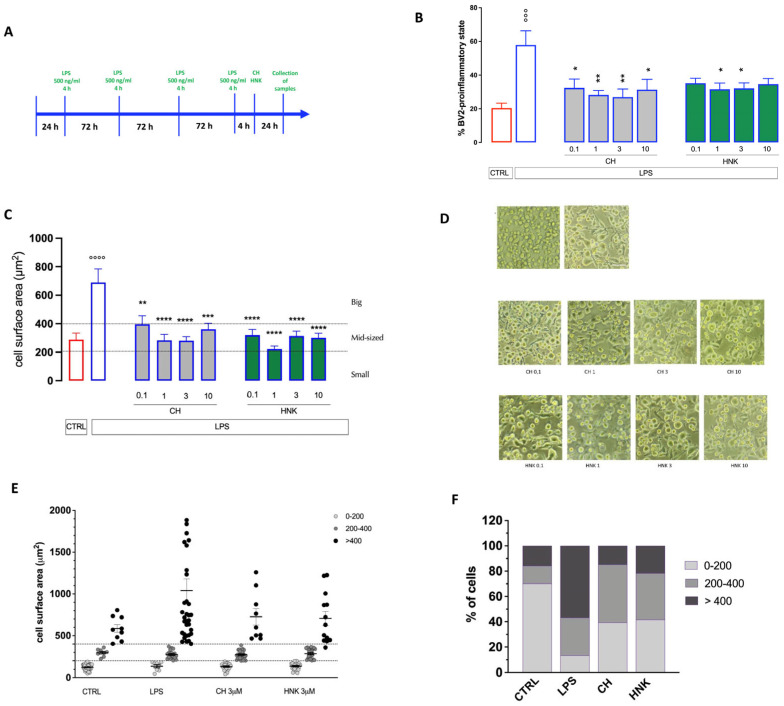
Effect of CH and HNK on the cell morphology of senescent BV2 cells. (**A**) Schematic representation of the experimental protocol. (**B**) Reduction in the percentage of BV2 cells in the pro-inflammatory state by CH and HNK (0.1–10 µM). (**C**) Reduction in the senescent BV2 cell surface area by CH and HNK (0.1–10 µM). (**D**) Representative images of CH- and HNK-treated senescent cells, bar 30 μm. (**E**) Scatter plot of redistribution of the senescent BV2 cell subpopulation by cell surface area after LPS intermittent stimulation in the presence or absence of CH and HNK. Cells were classified as either small (<200 µm^2^), mid-sized (200–400 µm^2^), or large (>400 µm^2^). (**F**) Effect of CH (3 µM) and HNK (3 µM) on LPS-induced variation in the percentage of distribution of small, mid-sized, and big cells. Three independent experiments were carried out to evaluate the effects of treatments. One-way ANOVA: °°° *p* < 0.001, °°°° *p* < 0.0001 vs. CTRL; * *p* < 0.05, ** *p* < 0.01, *** *p* < 0.001, **** *p* < 0.0001 vs. LPS.

**Figure 3 cells-13-01652-f003:**
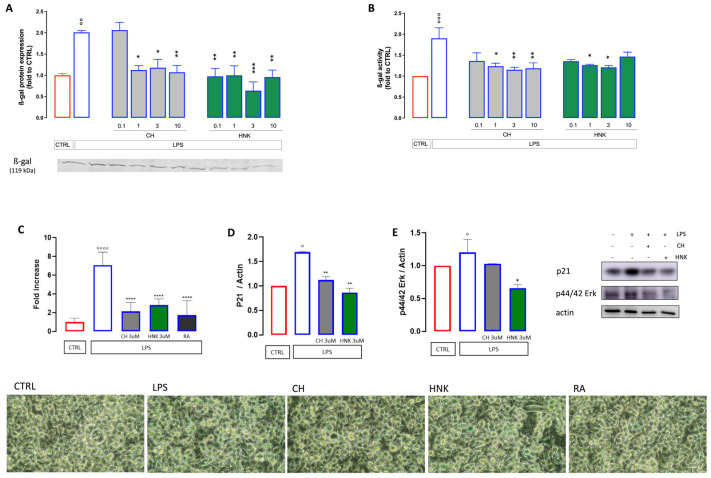
Effect of CH and HNK on cellular senescence markers. Intermittent stimulation with LPS at 500 ng/mL for 10 days increased both the protein expression (**A**) and activity (**B**) of ß-galactosidase in the cell lysate. Treatment with CH and HNK (0.1–10 µM) was effective in reducing both parameters. (**C**) Quantification analysis of ß-galactosidase staining after LPS stimulation and effect produced by CH and HNK (3 µM). RA (1 µM) was used as the reference drug. Representative images are reported at the bottom of the figure, bar 30 µm. One-way ANOVA: ° *p* < 0.05, °° *p* < 0.01, °°° *p* < 0.001, °°°° *p* < 0.0001 vs. CTRL; * *p* < 0.05, ** *p* < 0.01, *** *p* < 0.001, **** *p* < 0.0001 vs. LPS. Effect of LPS, CH, and HNK treatment on p21 (**D**) and p44/42 (**E**) expression level. Actin was used as the loading control. Three independent experiments were carried out to evaluate the effects of treatments. Representative blots are reported in the figure.

**Figure 4 cells-13-01652-f004:**
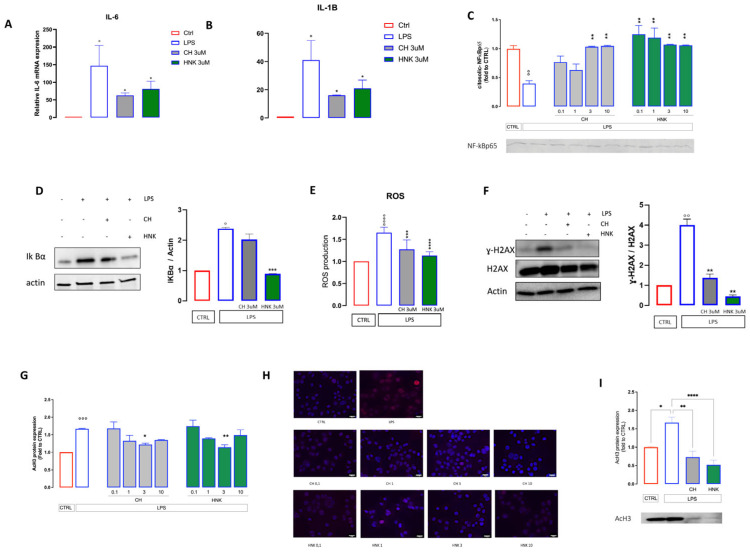
Anti-SASP activity of CH and HNK. (**A**–**B**) IL6 and IL1β mRNA levels in BV2 cells treated with LPS in the presence or absence of CH and HNK. Data are represented as fold change normalized to the mean expression of control (*n* = 3). One−way ANOVA: ° *p* < 0.05 vs. CTRL; * *p* < 0.05 vs. LPS. (**C**) CH and HNK (0.1–10 µM) restored the levels of cytosolic NF−kBp65 that were drastically reduced in senescent cells. °° *p* < 0.01 vs. CTRL; ** *p* < 0.01 vs. LPS. (**D**) The CH and HNK reduction of IKBα increased the expression induced by LP stimulation. Actin was used as the loading control. Western blot and the relative quantification are shown. (**E**) Effect of LPS, CH, and HNK treatment on total ROS production. The measurement was performed by incubating cells with an H2DCF-DA probe. Data were normalized on protein total content (*n* = 3). One-way ANOVA: °°° *p* < 0.001 vs. CTRL; *** *p* < 0.001 vs. LPS. (**F**) H2AX phosphorylation levels in BV2 cells stimulated with LPS in the presence or absence of CH and HNK. Actin and H2AX were used as the loading control (*n* = 3). Western blot and the γH2AX/H2AX ratio quantification are shown: ° *p* < 0.05,°° *p*< 0.01, °°° *p* < 0.001, °°°° *p* < 0.0001 vs. CTRL; ** *p* < 0.01 vs. LPS. (**G**) Reduction by CH and HNK (0.1–10 µM) of AcH3 increased the expression induced by intermittent LPS stimulation. (**H**) Representative images of CH- and HNK-treated LPS-stimulated cells stained with DAPI (nuclei) and AcH3, bar 20 µm. (**I**) The CH and HNK (3 µM) reduction of LPS-induced AcH3 increased expression and representative blot. * *p* < 0.5, ** *p* < 0.01, **** *p* < 0.0001.

**Figure 5 cells-13-01652-f005:**
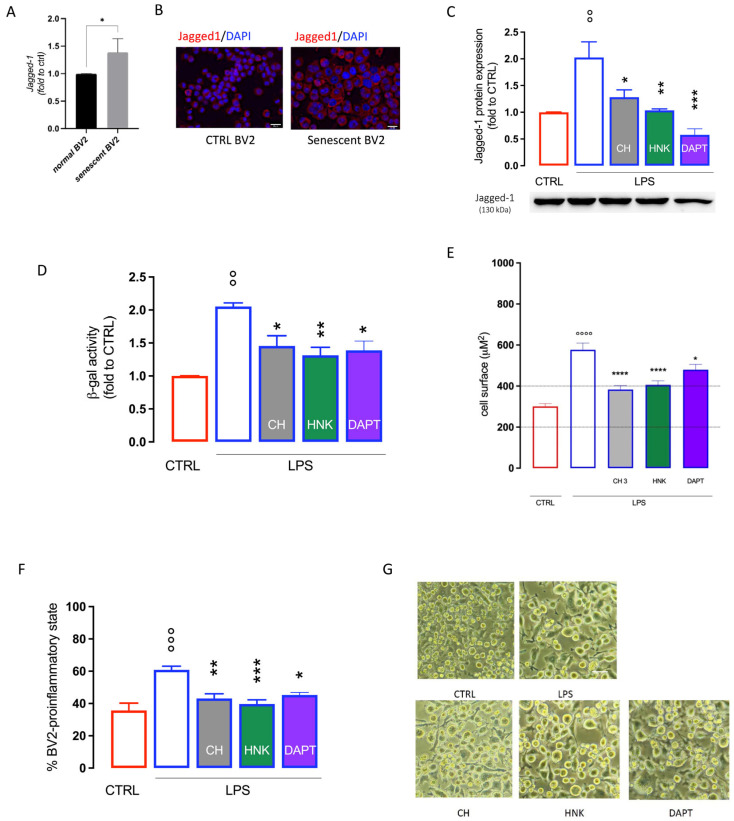
Effect of CH and HNK on Jagged-1 expression in BV2 senescent cells. Increase in Jagged-1 protein expression (**A**) and immunostaining (**B**) in LPS-stimulated senescent BV2 cells compared to unstimulated CTRL cells, bar 20 µm. (**C**) Reduction by CH and HNK (3 µM) of Jagged-1 increased expression in senescent BV2 cells. DAPT was used as a reference drug. (**D**) Attenuation of the ß-galactosidase (ß-gal) activity by CH, HNK, and DAPT. Reduction by CH and HNK of the cell surface area (**E**) and number of BV2 cells in the pro-inflammatory state (**H**). (**G**) Representative images of CH- and HNK-treated LPS-stimulated cells, bar 30 μm. * *p* < 0.05, ** *p* < 0.01, *** *p* < 0.001, **** *p* < 0.0001 vs. LPS; °° *p* < 0.01, °°° *p* < 0.001, °°°° *p* < 0.0001 vs. CTRL (one-way ANOVA).

**Table 1 cells-13-01652-t001:** Nucleotide sequences of primers.

*IL-6* (reverse)	5′-TGCATCATCGTTGTTCATAC-3′
*IL-6* (forward)	5′-CTTCCATCCAGTTGCCTTCT-3′
*IL-1β* (reverse)	5′-CCCATCAGAGGCAAGGAGGAA-3′
*IL-1β* (forward)	5′-CCTGCAGCTGGAGAGTGTGGA-3′
*β2M* (reverse)	5′-GTCATGCTTAACTCTGCAGG-3′
*β2M* (forward)	5′-TGCTATCCAGAAAACCCCTC-3′

## Data Availability

The data presented in this study are available on request from the corresponding authors.

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
