# Peer review of "The Neolignan Honokiol and Its Synthetic Derivative Honokiol Hexafluoro Reduce Neuroinflammation and Cellular Senescence in Microglia Cells"

_cells, 2024, doi:10.3390/cells13191652_

Round 1
Reviewer 1 Report
Comments and Suggestions for Authors
The article “The neolignan Honokiol and its synthetic derivative Honokiol Hexafluoro reduce neuroinflammation and cellular senescence in microglia cells” presents new biological properties of the tested substances (act as senomorphic drugs) and at the same time indicates their potential as modulators of novel options for neurodegenerative disease treatment targeting the microglia cells (neuroinflammation and senescence). I believe the work has enough novelty to be published in "Cells.”.
The aim was formulated correctly, and the presented results confirm the thesis. However, the article requires some minor corrections before publication.
Introduction:
The plant name should be written in italics.
(Zhang et al., 2024) -> [23]
Material and methods:
2.1. BV2 cell culture: (CO2) should be changed to: CO2
2.2. Drug administration: how long were the cells incubated with the tested substances? 24/48/72/96 h? It is hard to find this information.
2.3. Sulforhodamine B (SRB) Assay- 200ml à probably should be 200μM
Everywhere: “rt” -> RT (capital letters) unify throughout the article.
2.9. Real-Time PCR (RT-PCR): “retro-transcribed” -> reverse...
I recommend putting primer sequences into tables. It will be better readable. The genes' names should be written in italics.
Results:
3.4 Anti-SASP activity of CH and HNK: Figure 4D – vinculin or actin?
Is it possible to get the figures in better resolution?
Besides some small, needed corrections and minor revisions, I recommend the article be published.
Author Response
We thank the reviewer for his/her positive comments on our manuscript. We have now added the suggested minor corrections.
Q:Introduction:
The plant name should be written in italics.
R:Done (Pagg 1 and 2)
Q:(Zhang et al., 2024) -> [23]
R:Done (now ref 22) (pag 2)
Q:Material and methods:
2.1. BV2 cell culture: (CO2) should be changed to: CO2
R:Done (pag 2)
Q: 2.2. Drug administration: how long were the cells incubated with the tested substances? 24/48/72/96 h? It is hard to find this information.
R: As reported in the diagram in Fig. 2A, cells were incubated with tested substances for 24 h. We now added this information to the method section, too (pag 3).
Q: 2.3. Sulforhodamine B (SRB) Assay- 200ml probably should be 200μM
R: We apologize for the typo. 200 ml should be 200 µl and refers to the amount of acetic acid added to the well. The correct sentence is “wells were washed four times with 1% acetic acid (200 µl per well) and 200 µl of TRIS HCl solution (pH 10) were added to the wells and incubated for 5 min with shaking (pag 3)”
Q: Everywhere: “rt” -> RT (capital letters) unify throughout the article.
R: Done (pag 3)
Q: 2.9. Real-Time PCR (RT-PCR): “retro-transcribed” -> reverse ...
R: Done (pag 4)
Q: I recommend putting primer sequences into tables. It will be better readable. The genes' names should be written in italics.
R: We wrote the genes’ name in italics and we inserted the primers sequence into a table (pag 4)
Q: Results:
3.4 Anti-SASP activity of CH and HNK: Figure 4D – vinculin or actin?
R: We apologize for the inaccuracy: actin is the normalizer, we changed the y-axis of the bargraph, accordingly. (pag 9)
Q: Is it possible to get the figures in better resolution?
R: In the revised version of the manuscript figures in higher resolution have been inserted inserted.
Besides some small, needed corrections and minor revisions, I recommend the article be published.
Reviewer 2 Report
Comments and Suggestions for Authors
The authors illustrate a protective activity of HNK and CH on neuroinflammation and cellular senescence in microglia cells involving a Notch signaling-mediated mechanism and suggesting a potential therapeutic contribution in aging-related neurodegenerative diseases. I have follow suggestions;
Not a strong rationale presented to study HNK for aging. Add few lines in introduction.
Why were mouse cell line used. this study could have been performed in human microglial cell line.
500ng seems too toxic for cells? what is the rationale of killing cells by LPS. Why not just observe the inflammatory response after LPS treatment. More the survival of the cells more the activation and neuroinfalmmation. this controversy needs to be addressed.
"Drug administration"; the heading is misleading. it looks drug is administered to animals.
the image quality is low.
the bar in the imaging pictures are not shown consistent.
No attempt is made to model senesce in microglial cells.
Author Response
We thank the reviewer for his/her suggestions on our manuscript. We have now modified the text, accordingly:
Q: Not a strong rationale presented to study HNK for aging. Add few lines in introduction.
R: To better highlight the rational for studying HNK for aging, this comment has been inserted in the introduction: “In the research of new senotherapeutics, natural phenolic compounds, usually endowed with high efficacy and tolerability, represented an interesting perspective. Since senescent cells propagate systemic inflammation through activation of pro-oxidant and proinflammatory pathways, polyphenols, molecules endowed with antioxidant and anti-inflammatory activity, might hold promise as “anti-senescence” approach.” (pag. 2)
Q: Why were mouse cell line used. this study could have been performed in human microglial cell line.
R: LPS-stimulated BV2 cells represent a well-studied and reliable model of neuroinflammation. Recently, intermittent stimulation of BV2 with LPS for 10 days has been characterized as a reliable model of microglia senescence (Yu et al., 2012; Borgonetti and Galeotti 2022) that recapitulate main features of senescent cells in rodents (Muralidharan et al., 2022; Borgonetti and Galeotti, 2023). Thus, to better validate our results we focused on this model, and we tested HNK and CH on BV2 senescent cells. However, the lack of data on human microglia cells represents a limitation and the next step will be to better investigate the activity of HNK and derivatives in other microglia cell lines.
Q: 500 ng seems too toxic for cells? what is the rationale of killing cells by LPS. Why not just observe the inflammatory response after LPS treatment. More the survival of the cells more the activation and neuroinfalmmation. this controversy needs to be addressed.
R: Irreversible cell cycle arrest is the main feature of senescent cells. Excessive and aberrant accumulation of senescent cells in tissues negatively affects regenerative capacities. Simultaneously, this condition creates a proinflammatory milieu favorable for the onset and progression of various age-related diseases. Thus, the proinflammatory response of senescent cells is accompanied by a reduction in cell number which, along with other typical traits of cellular senescence such as increased ß-gal activity, SAHF, p38 and ERK expression, define the status of senescent cell (Borgonetti and Galeotti, 2022;2023). All these markers are detectable using LPS 500 ng while lower doses promote inflammation without inducing cellular senescence, thus representing a model of neuroinflammation. For this reason, we stimulated BV2 with LPS 250 ng for 24 h to induce neuroinflammation or with LSP 500 ng intermittently for 10 days to induce cellular senescence.
Q: "Drug administration"; the heading is misleading. it looks drug is administered to animals.
R: We substituted “drug administration” with “Cell treatment” (pag 3)
Q: the image quality is low.
R: We inserted figures in higher resolution in the revised version of the manuscript.
Q: the bar in the imaging pictures are not shown consistent.
R: We now added the bars to the imaging pictures (see Fig 1C, 1E, 2D, 3 and 5G). The bars in Fig 5A and 4H which were already present in the previous version of the manuscript are now more evident due to the higher resolution of the images.
Q: No attempt is made to model senesce in microglial cells.
R: The cellular senescence in microglia cells has been modelled and characterized previously by several studies (Yu et al., 2012; Borgonetti and Galeotti 2022; 2023). To avoid both repetition of published data and lengthening the manuscript, we prefer not to include this information in the present manuscript.
Reviewer 3 Report
Comments and Suggestions for Authors
The title of the present work, "The Neolignan Honokiol and its Synthetic Derivative Honokiol Hexafluoro Reduce Neuroinflammation and Cellular Senescence in Microglia Cells," highlights the protective effects of two compounds, HNK and CH, on inflammation and microglial senescence. The authors present a well-supported potential mechanism of action for these compounds; however, the manuscript requires further improvement before it can be considered for publication.
Minor revision:
- The authors should revise the citation format, most of them are numbers but there is one with another format.
- Authors should mention the number of experiments performed in every method section.
- Why did the authors lyse the cells for the total ROS production instead of reading the fluorescence directly in the plate reader?
- Could the authors clarify why certain control bars are presented without error bars?
- In section 3.1 about the effect of HNK and CH in an in vitro model of neuroinflammation, why do the authors talk about senescence induction (second paragraph)?
- Figure 1C is not described in the results section.
- Figure 1B, Is the RA group significantly different with respect to the LPS group?
- Figure 1I, the CH 0.1 and 1 bars are not significantly different from LPS group, is that correct?
- In Figures 2E and F, the authors should use the same colours to represent the graph as the graph 1G and H.
- The sentence in section 3.3, 'Quantification of SA-β-gal staining further confirmed the increased SA-β-gal activity in intact cells and the capability of CH and HNK to restore activity to control values,' is unclear and seems to convey a different message than what is represented in the graph.
- The figure legend for Figure 3 is poorly explained. Does Figure 3C correspond to the quantification of the bands or the expression/activity of β-gal?"
- In Figure 3E, the authors claim that similar results were obtained for this marker however, one of the compounds is not significantly different with respect to LPS group. Please, mention that in the text.
- In general, the figures are quite small and of poor quality.
- Could the authors clarify the use of different housekeeping proteins/Ponceau for the western blot analysis?
- Figures 4A and B, are the compound groups significantly different to the LPS group? It is not represented in the figure but claimed in the text. Please, clarify this.
- Figure 4C, did the authors perform cell fractionation to measure NF KB levels in the cytosol? Could the authors explain this?
- Figure 4D, which housekeeping protein is used actin or vinculin?
- In the results for Figure 4E, F, the authors stated, “and decreases the expression of H2AX phosphorylation”, but they should say “decreases the phosphorylation levels of H2AX”.
- Figure 4G, H, why immunofluorescence was used to quantify the expression of acH3 instead of western blot?
- In the results section, “HNK and CH (3 μM) reduced SA-ß-gal expression (Fig. 5C)”, figure 5C does not correspond to SA-ß-gal expression and figure 5D represents the activity. Please, clarify this.
- The authors should represent western blot bands for all western blot figures.
Major revision:
- To study phosphorylated proteins, it is essential to detect the total levels of the target protein and present the phosphorylation ratio. The authors should do this to ensure accurate results
- The authors should provide western blot analysis to quantify the expression of acH3.
Comments on the Quality of English LanguageThe quality of the English language is correct.
Author Response
We thank the reviewer for his/her suggestions on our manuscript. We have now modified the text, accordingly:
Minor revision:
Q;:The authors should revise the citation format, most of them are numbers but there is one with another format.
R: Citation format has been revised (now ref 22, pag 2).
Q: Authors should mention the number of experiments performed in every method section.
R: The number of independent experiments performed has been added to each method section and in the figure legends (pagg 3-4 and pagg 6-8).
Q: Why did the authors lyse the cells for the total ROS production instead of reading the fluorescence directly in the plate reader?
R: We currently lyse cells and then read fluorescence according to published protocol for ROS quantification with DCFDA probe using a fluorescent microplate reader (Hyeoncheol Kim and Xiang Xue PMID: 32658187) and according to previous results obtained in our lab (Ippolito L et al., PMID: 27542265; Pardella E PMID: 36552790) However, we agree with the reviewer that it is possible to detect DCFDA also on living cells as we performed in Taddei ML et al by confocal analysis (PMID: 17280488). For experimental reproducibility, we currently use lysed cells for detection fluorescence with a microplate reader and live cells for confocal and/or cytometry analysis. No particular differences in sensitivity have been observed.
Q: Could the authors clarify why certain control bars are presented without error bars?
R: In some control values the s.e.m. value is too small to be visible as error bar.
Q: In section 3.1 about the effect of HNK and CH in an in vitro model of neuroinflammation, why do the authors talk about senescence induction (second paragraph)?
R: We agree with the reviewer that in the in vitro model of neuroinflammation, it is not correct to talk about “senescence induction” since we are observing modifications in cell morphology induced by LPS treatment, thus we modified the text accordingly (pag. 5). In addition, the model of neuroinflammation and the model of cellular senescence are different and in the second paragraph we refer to data about senescent microglia cells. To avoid any misunderstanding, we re-numbered the paragraphs in the Results section (pag 6).
Q: Figure 1C is not described in the results section.
R: Fig. 1C has been inserted in the results section (pag 5).
Q: Figure 1B, Is the RA group significantly different with respect to the LPS group?
R: Yes, it is. We revised the figure by adding the appropriate asterisks (pag 6).
Q: Figure 1I, the CH 0.1 and 1 bars are not significantly different from LPS group, is that correct?
R: Yes, both CH values are not significantly different from LPS.
Q: In Figures 2E and F, the authors should use the same colours to represent the graph as the graph 1G and H.
R: In Fig. 1 we reported the results from the model of neuroinflammation while Fig. 2 reports data from senescent cells. We used different colours to better emphasize the difference between the two models. In agreement with the reviewer’s suggestion, to improve consistency, we edited figures by using the same colours (pagg 6 and 7).
Q: The sentence in section 3.3, 'Quantification of SA-β-gal staining further confirmed the increased SA-β-gal activity in intact cells and the capability of CH and HNK to restore activity to control values,' is unclear and seems to convey a different message than what is represented in the graph.
R: The SA-β-gal activity was measured in cell lysates while SA-β-gal staining was performed in intact cells. We inserted in the manuscript images and quantification of SA-β-gal staining to further validate values obtained with a standard method of SA-β-gal activity quantification that is performed in cell lysates. Similar activities for HNK and CH were obtained. We rephrase the sentence for better clarity (pag 8).
Q: The figure legend for Figure 3 is poorly explained. Does Figure 3C correspond to the quantification of the bands or the expression/activity of β-gal?"
R: Legend to Fig. 3 has been revised (pag 8). Fig. 3C corresponds to quantification of β-gal staining.
Q: In Figure 3E, the authors claim that similar results were obtained for this marker however, one of the compounds is not significantly different with respect to LPS group. Please, mention that in the text.
R: The comment on results from Fig. 3E has been revised (pag 8).
Q: In general, the figures are quite small and of poor quality.
R: We inserted figures in higher resolution in the revised version of the manuscript.
Q: Could the authors clarify the use of different housekeeping proteins/Ponceau for the western blot analysis?
We normalized values for IL-10 and Jagged to the total protein content since cytokines and Notch pathway mediators are under control of some RNA binding proteins (i.e. ELAVs) that are expressed by microglia cells. Among the numerous mRNA targets, these proteins might control the expression of ß-actin. Thus, to avoid potential misinterpretation of data, we reported data normalized by Ponceau staining.
Q: Figures 4A and B, are the compound groups significantly different to the LPS group? It is not represented in the figure but claimed in the text. Please, clarify this.
R: Yes, they are. We missed the appropriate asterisks in the compound groups in the previous version of the manuscript, we now inserted the asterisks and the statistical analysis in fig 4A and 4B (pag 9).
Q: Figure 4C, did the authors perform cell fractionation to measure NF KB levels in the cytosol? Could the authors explain this?
R: Cytosolic NF-kB levels were measured in the cytosolic fraction after cell fractionation. Cell fractionation procedure has now been added in the M&M section (pag 4).
Q: Figure 4D, which housekeeping protein is used actin or vinculin?
R: We apologize for the inaccuracy: actin is the normalizer, we changed the y-axis of the bargraph, accordingly (pag 9).
Q: In the results for Figure 4E, F, the authors stated, “and decreases the expression of H2AX phosphorylation”, but they should say “decreases the phosphorylation levels of H2AX”.
R: We apologize, now the text has been corrected (pag 9).
Q: Figure 4G, H, why immunofluorescence was used to quantify the expression of acH3 instead of western blot?
R: We performed immunofluorescence experiments to define the cellular localization of AcH3 to correlate this marker to cellular senescence. Immunofluorescence staining of images can also be quantified through widely used standard methods routinely used in our laboratory that produce results comparable to those obtained by western blotting. In the original version we preferred to quantify the same images instead of performing western blot experiments to minimize any potential experimental bias originated from different experimentations on different biological samples. Accordingly to the reviewer suggestion, we have now inserted results from western blotting experiments performed to quantify AcH3 protein levels (Fig. 4I) that showed similar results to those obtained from IF staining quantification.
Q: In the results section, “HNK and CH (3 μM) reduced SA-ß-gal expression (Fig. 5C)”, figure 5C does not correspond to SA-ß-gal expression and figure 5D represents the activity. Please, clarify this.
R: We apologize for the typo. The text has been corrected (pag 10).
Q: The authors should represent western blot bands for all western blot figures.
R: We inserted the blot band for IL10 in Fig.1 (pag 9) that, in the original version, was reported by mistake only as supplementary material in the uncropped blots file.
Major revision:
Q:To study phosphorylated proteins, it is essential to detect the total levels of the target protein and present the phosphorylation ratio. The authors should do this to ensure accurate results
R: We thank the reviewer for the observation, and we agree with her/him that the best way to study phosphorylation is to normalize for the target protein. For what concern p42/p44, we actually quantified the expression level of the proteins that increases following LPS treatment while the two compounds are able to revert this effect. We replaced the word “activation”, that was misleading, with “expression”, a much more appropriate term for this result (pag 8). In agreement with the suggestion, for what concern the phosphorylation of H2AX, we performed a western blot to detect the total level of H2AX and now we present the results in the bargraph as phosphorylation ratio (pag 9).
Q: The authors should provide western blot analysis to quantify the expression of acH3.
R: To further confirm data from IF staining quantification of AcH3, we here report results for AcH3 protein levels quantification obtained by WB analysis (Fig. 4I).
Round 2
Reviewer 2 Report
Comments and Suggestions for Authors
All comments have been addressed
Reviewer 3 Report
Comments and Suggestions for Authors
Thank you to the authors for addressing all the questions and concerns raised during the review process. I believe this version of the manuscript is now suitable for publication in this journal.